# Natural Gas Induced Vegetation Stress Identification and Discrimination from Hyperspectral Imaging for Pipeline Leakage Detection

**Pengfei Ma** , **Ying Zhuo, Genda Chen \*** and **Joel G. Burken**

Department of Civil, Architectural and Environmental Engineering, Missouri University of Science and Technology, Rolla, MO 65401, USA; pm7m8@mst.edu (P.M.); yingzhuo@mst.edu (Y.Z.); burken@mst.edu (J.G.B.)
\* Correspondence: gchen@mst.edu

**Abstract:** Remote sensing detection of natural gas leaks remains challenging when using ground vegetation stress to detect underground pipeline leaks. Other natural stressors may co-present and complicate gas leak detection. This study explores the feasibility of identifying and distinguishing gas-induced stress from other natural stresses by analyzing the hyperspectral reflectance of vegetation. The effectiveness of this discrimination is assessed across three distinct spectral ranges (VNIR, SWIR, and Full spectra). Greenhouse experiments subjected three plant species to controlled environmental stressors, including gas leakage, salinity impact, heavy-metal contamination, and drought exposure. Spectral curves obtained from the experiments underwent preprocessing techniques such as standard normal variate, first-order derivative, and second-order derivative. Principal component analysis was then employed to reduce dimensionality in the spectral feature space, facilitating input for linear/quadratic discriminant analysis (LDA/QDA) to identify and discriminate gas leaks. Results demonstrate an average accuracy of 80% in identifying gas-stressed plants from unstressed ones using LDA. Gas leakage can be discriminated from scenarios involving a single distracting stressor with an accuracy ranging from 76.4% to 84.6%, with drought treatment proving the most successful. Notably, first-order derivative processing of VNIR spectra yields the highest accuracy in gas leakage detection.

**Keywords:** remote sensing; hyperspectral imaging; vegetation stress; methane/natural gas; pipeline leakage detection; multivariate analysis; climate change





## 1. Introduction

Today, the global demand for natural gas is soaring high. Thus, it becomes increasingly important to minimize any unintended release of natural gas for operational safety, economic benefits, and control of climate change, as methane is the second largest contributor to global warming. Over the last decade, the estimated anthropogenic gas emission reached 359 Tg [1] and underground pipelines were susceptible to gas leakage when damaged under man-made and natural hazards. Between 2003 and 2022, over 12,781 pipeline incidents occurred in the US [2]. These incidents were caused by natural forces (earth movement, wind gusts, heavy rains/floods, lighting), excavations from third parties or operators, operation negligence, material defects, and corrosion [3]. The detection of gas leakage from pipelines has long been viewed as a challenging task due to its unpredictable leakage scale, especially in vegetated regions with limited access to potential accident locations [4]. The undiscovered gas leakage may result in severe environmental hazards like greenhouse emissions and even cause fatality to both humans and animals due to its volatility. Remote sensing can overcome the access limit and help rapidly screen vegetation-rich ground conditions. This technology makes it possible to detect vegetation alterations associated with the gas leakage, though its efficiency largely depends upon the use of sensors [5–9].

Unmanned aircraft vehicles (UAVs) enable remote sensing that can be used to timely detect the conditions of plants through physical and biochemical characteristics. Gas leak-

age depletes the oxygen in the soil and alters the microbiological conditions to discourage plant root respirations (energy supply) in hypoxic conditions, thus reportedly stressing plants [10–12]. Thermal imaging has been introduced to detect gas-stressed plants because their stomatal activity deviated from corresponding unstressed conditions, forming a temperature difference in the plant canopy [13]. RGB imaging can also capture the color alteration of the plants associated with gas leakage [14]. The two techniques were combined to detect the plant stress associated with water deficit [5]. However, thermal imaging detection is highly dependent on temperature resolution and sensitivity because the stressed plants exhibit an insignificant temperature difference from that of the normal plants. RGB image analysis to spot gas leakage generally employs the visible symptoms on plant foliage and the reduced growth of impacted vegetation over time, which cannot meet the need for short-term detection.

Hyperspectral imaging (HSI) is a different remote sensing technique for the identification of plant stress because it enables the spectroscopic analysis of physiological and biochemical changes at pixel, canopy, and plant levels [15–18]. The reflectance curve (spectral signature/curve) retrieved from hyperspectral imaging is an indication of the interaction between leaf chemical compositions and electromagnetic radiation. The spectral signature is a function of wavelengths ranging from visible (VIS, 400–750 nm), near-infrared (NIR, 750–1200 nm), to short-wave infrared (SWIR, 1200–2400 nm) [19–21]. Chemical compounds of the plants are represented by various features on a spectral curve. For example, $H_2O$ yields absorption valleys at 1410 nm and 1930 nm due to the presence of O-H [22–24]. Exposure to adverse conditions stimulates the synthesis of antioxidants to counteract reactive oxygen species (ROS), such as radicals OH•− and OH• and non-radicals like $H_2O_2$, and thus protects the plants from the toxic, damaging impacts [25]. Plant stress is a physiological product of plant pigments, proteins, and cell structure component alterations. Photosynthetic pigmentations dominate VIS light scattering over the parenchyma structure. Outer-layer epidermis affects the radiation absorption in NIR, while enzyme (protein), lignin, cellulose, and water content are correlated with the SWIR spectroscopy. Therefore, hyperspectral imaging responds to the changes on leaf surfaces through the features of spectral signatures with stress response at specific wavelengths [26,27].

Gas leakage detection with HSI data has been involved in several studies. Vegetation indices (VIs) in ratio have been used to effectively associate band information with specific biochemical parameters; thus, the gas indued stress is present on plants. VIs are equations developed to represent specific changes in reflectance at particular wavelengths compared to a reference wavelength. VIs are proxies of the plant health status assessed by an increase or decrease of the index's value. For instance, Normalized Difference Vegetation Index (NDVI) increases represent healthier plants (more greenness), while the increase in the Anthocyanin Reflectance Index (ARI) can indicate plant stress [28]. A natural gas indicator constructed by Ran et al. (2020) can demonstrate the underground gas leak when the Jeffries–Matusita (JM) distance exceeds 1.8 [29]. In addition, Pan et al. (2022) compared several gas stress indicators and proposed a newly defined variational mode decomposition index (VMDI), which outperforms other indicators in terms of early detection [30].

Different plant stressors can impact the reflectance in hyperspectral curves at specific wavelengths. The reflectance ratio between 725 nm and 702 nm (R725/R702) decreased by 30% to 50% before any visible signs could be observed on tested grasses under increasing gas stress and the normalized difference index between 750 nm and 1900 nm gave a gas leakage detection with high sensitivity [31]. Accettura (2018) predicted the developmental stage of gas stress on maize and wheat with six reflectance indices. A combination of 760 nm, 1650 nm, and 1790 nm bands yields a detection of methane stress with 93% accuracy [32]. An alternative band combination of 760 nm, 790 nm, 820 nm, 880 nm, and 930 nm exhibits similar accuracy. Noomen et al. (2012) tested four established stress indicators for their effectiveness in mapping hydrocarbon seepages and revealed that the Lichtenthaler index (R440/R740) can reduce false abnormalities [10]. Resembling VIs, the 'red edge' is considered the most effective indicator of gas leakage stress on vegetation,

both in pixel and canopy scale [16,21,33]. Moreover, the 'red edge position' (REP), defined as the inflection point of at the reflectance spectrum between 680 nm and 750 nm, shifted to shorter wavelengths (also known as 'blue shift') and distinguished gassed vegetations due to the degradation of chlorophyll [33–35]. Although VIs are promising to discern gas-induced plant stress from other effects, the identification of gas-stressed plants can be misleading with false alarms because VIs incorporate information only from several wavelengths. Loss of information biases gas detection from other stressors unless the unique metabolic response due to the gas exposure in the rhizosphere can be ascertained. Note that REP impacts can be observed in other abiotic scenarios like salinization, heavy metal contamination, and water deficit attack [33,36–38].

Rather than VIs, the use of a full spectrum in the detection of gas leakage improves robustness and accuracy as physiological, morphological, and biochemical changes are considered simultaneously, though close-range wavelengths are somewhat interrelated, causing redundant messages in spectral analysis [39]. Moghimi et al. (2018) compared pixel similarity with vector-wise similarity to a salt endmember and identified different concentrations of salt treatment from similar indices [40]. Mirzaei et al. (2019) introduced a partial least square (PLS) method to group different grapevine foliage contaminated by heavy metal salts based on the NIR spectra [41]. Asaari et al. (2018) confirmed the effectiveness of full-spectrum identification for the early drought stress on maize plants cultivated in a high-throughput plant phenotyping platform with the analysis of variance (ANOVA) [19]. Differentiation between stressors has also been investigated from an integration of entire obtained spectra. From the reflectance curve of vegetation, Lassalle et al. (2018) assessed oil-contaminated soils in the presence of heavy metal with an accuracy of 89% and 93% from changes in the leaf and canopy of tested plants, respectively [42].

The above reviews indicated no systematic investigations on the selection of spectral ranges for the effective detection of vegetation stress. More importantly, the detection of gas-induced vegetation stress remains unexplored for specific wavelength responses and potential responses to existing VIs or the development of new, gas-stress-specific VIs. The impact of elevated concentrations of $CO_2$ in the root zone was conducted on the normalized difference first derivative index (NFDI), chlorophyll normalized difference vegetation index (Chl NDI), and other investigated VIs [43].

Gas pipelines were distributed across the boundaries of states or countries in various environments where other natural stressors such as salinity impact (SI), heavy metal contamination (HMC), and drought exposure (DE) are likely mixed with the potential effect of gas leakage in different ways. The other natural stressors can be severe disturbances to the detection of gas-induced stress because of their potential overlapped hyperspectral response on spectral signatures [44]. The multiple presence of stressors gives rise to false alarms in gas leakage [12,31,45,46]. To facilitate rapid and accurate detection of gas leakage with UAVs-enabled hyperspectral imaging, it is necessary to discriminate gas-stressed vegetation from vegetation displaying responses to natural stressors. Even with some false negatives, remote detection can expedite the deployment of 'on the ground' testing for affirming and mitigating potential leaks and eliminating false detections. To this end, three research questions need to be answered:

(1) Do hyperspectral curves respond to spectral features at unique wavelength bands for different stressors?
(2) Do plants display significantly different sensitivities to various stressors in VIS, NIR, and SWIR ranges during the identification and discrimination of gas-stressed vegetation?
(3) Are plant responses (e.g., metabolic response) similar across different species so that specific vegetative indices may be used for many sites and vegetations?

To reduce the number of false alarms when detecting gas leakage from hyperspectral reflectance, three species of plants, one grass and two shrubs, will be tested in a greenhouse under three natural stressors: SI, HMC, and DE. They will be imaged over time using a hyperspectral camera. The research objectives of this study are:

1. To develop an accurate and robust approach for gas leakage identification from the change of hyperspectral reflectance spectra instead of VIs,
2. To develop an effective method to spectrally discriminate gas leakage from the other possible concurrent natural stressors, and
3. To determine the spectral region(s) that yield the most accurate classification of gas leakage from trained discriminants in multivariate analysis.

## 2. Materials and Methods

### 2.1. Lab Test Design

Greenhouse tests were conducted in the Hypoint Laboratory (37.955376°N, 91.771681°W) at Missouri University of Science and Technology (S&T). As shown in Figure 1, one grass (*Calamagrostis* × acutifoliate Karl Foerster grass abbreviated as 'Grass') and two shrubs (*Ligustrum sinense* 'southern sunshine' abbreviated as 'South' and *Ilex glabra* Gem box® inkberry holly abbreviated as 'Gem') were selected to emulate gas leakage effects because of their presence in the wild of North America. All plants were mature to ensure that their heights did not change appreciably in the study period, having minimal influence on subsequent hyperspectral scanning. The selected plants were perennial to overcome aging deterioration. The plants were treated with methane gas and three other stressors: SI, HMC, and DE. The other stressors served as distraction sets for gas leakage detection [47–49]. For comparison, plants cultivated under optimal conditions without any known stress were used as a control reference, which is referred to as a non-stressed scenario. For each stress treatment condition and the non-stressed scenario, three tests were repeated. Therefore, a total of 45 pots of plants (3 replicas × 5 treatments × 3 species) were prepared for the greenhouse tests.

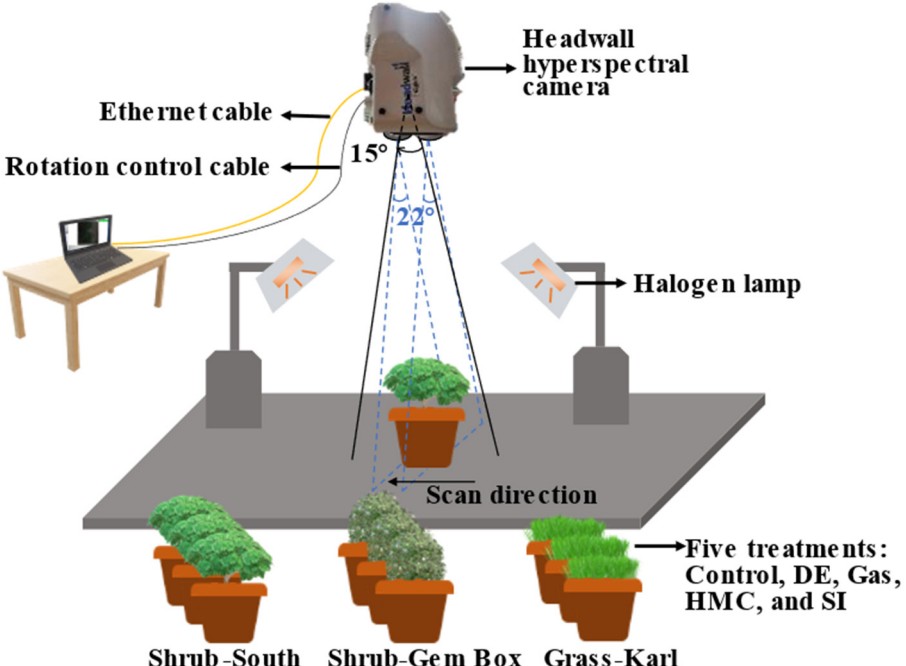

**Figure 1.** Schematic demonstration of the regular hyperspectral imaging collection on vegetations with a hyperspectral camera in lab settings.

### 2.2. Stress Treatments

Before any treatment, all plants were placed in the greenhouse for an acclimation period of 15 days to ensure they were adapted to the greenhouse environment. For gas treatments, plants were transplanted to 8-L standard cylinder pots for easy distribution of methane gas. A percolated gas distributor in cross shape was installed at the bottom of each pot for methane application. The four ends of the cross were alternated clockwise

to ensure that methane diffuses uniformly into the soil. Ultra-high purity grade methane from Airgas (Airgas Inc., Radnor, PA, USA) was used as a stress medium to stimulate the plants. The flow rate of the methane was regulated to 5 L/h over 10 h a day. The gas was delivered through transparent vinyl tubing (D = 0.1875 cm). For SI treatments, the soil was amended with NaCl and CaCl$_2$ in 2:1 mol ratio to realize a moderate salinity for each species of plants [50]. Here, sodium and calcium chloride salts were used because of their abundance in nature. The moderate salinity was quantified by the soil-saturated paste electrical conductivity (*ECe*) [51]. In reference to saline resistances of different species, *ECe* was finally set to 6 dS/m, 8 dS/m, and 8 dS/m for Grass, South, and Gem, respectively [52]. Before salinization, the original *ECe* of soil in each pot of plant was measured and the amount of salt needed was estimated. For HMC treatments, the composition of heavy metal elements and their concentration in soil were referred to the U.S. Department of Agriculture (USDA) regulatory limits, as given in Table 1. In this study, the five most common heavy metal elements are considered: chromium, copper, zinc, nickel, and arsenic. Heavy metal salts are diluted into irrigation water and sprayed on the soil of potted plants in three batches to prevent overflowing [42]. For DE treatments, irrigation water was reduced to half of the water intake than normal [12,21]. After each treatment, plants were transferred back to the greenhouse to mark the start of a stress cycle. For the reference group (Ref), plants were watered as instructed without any additional treatment on the soil to create a stress-free environment.

**Table 1.** USDA regulatory limits of heavy metals applied to soil developed by the U.S. Environmental Protection Agency (EPA).

| Heavy Metal Salt | As | Cd | Cr | Cu | Pb | Hg | Ni | Se | Zn |
|---|---|---|---|---|---|---|---|---|---|
| Maximum (ppm) | 75 | 85 | 3000 | 4300 | 420 | 840 | 75 | 100 | 7500 |

The test lasted 121 days and all the plants were cultivated in the greenhouse with a temperature of 25 °C and a relative humidity of 70%. A series of LED lights with a constant 378 μmol·m$^2$/s photosynthetically active light intensity were uniformly distributed on the roof of the greenhouse to create a homogeneous radiation on each plant as it was in a normal transpiration system. A photoperiod of 14 h of light and 10 h of darkness was maintained throughout the greenhouse tests. In addition, the ventilation rate of the greenhouse was set to 50 L/min to achieve good air circulation for the plants in the greenhouse and maintain a constant temperature.

*2.3. Hyperspectral Imaging and Calibration*

2.3.1. Hyperspectral Camera Setup

The stress effects on the treated plants were characterized by hyperspectral reflectance, also known as spectral signature or spectral curve. A push broom hyperspectral imaging platform built for this test was composed of four parts: imager, spectrometer, illumination source, and movement control, as shown in Figure 1. The Headwall dual-lens imager (Headwall Photonics, Inc., Bolton, MA, USA) that covers a full spectral range from 400 nm to 2400 nm was used to collect hyperspectral images. The full spectral range can be divided into two regions: VNIR (400–1000 nm) and SWIR (1000–2400 nm). The VNIR range has 249 bands with 2 nm spectral resolution, while the SWIR range has 240 bands at a 6 nm wavelength interval. Therefore, the full spectrum has 489 sampling bands. The external illumination was provided by two 300 W full-spectrum tungsten halogen lamps (Ushio Lighting Inc., Cypress, CA, USA) that were arranged parallel with the plant pots to produce homogeneous radiation on the canopy of plants. Under constant light exposure, the exposure time of the hyperspectral camera was set to 40 ms at a framing rate of 45 ms. The field of view of the camera on the side of two lenses was 22°. In the plane of the dual lens, the hyperspectral camera can rotate 15° to cover the area to be scanned during tests. The speed of the hyperspectral camera was adjusted to fit the imaging setup to ensure no

pixel distortion so that the object in the pixel was neither compressed nor extended in the movement direction. The camera parameter setup and speed control were finished in the software Hyperspec III (Version: 3.1, Headwall Photonics, Inc., Bolton, MA, USA).

To collect hyperspectral images, the plant canopy was set to be 1.2 m down below the imager. For each scanning, a screening line contains 640 pixels within the Field of View (FOV) of the imager and the obtained image has 640 × 1208 pixels. The spatial resolution of the pixel is 0.78 mm. Before the plant canopy was scanned, a gray mat was placed underneath each pot within the view of the camera to reduce the reflection from the background, lowering the risk of shadowing in the hyperspectral image. All the plants were scanned every five days after the acclimation period.

### 2.3.2. Radiometric Calibration

Hyperspectral reflectance extracted from the plant leaves reflects the physical, morphological, and biochemical status of the leaf surface [37]. The quality of image data is related to the image detector properties such as lens, sensor, grating, and filter. Raw images are subjected to radiometric calibration to reduce the influence of variability from the sensor [53,54]. Raw digital numbers (DNs) captured by the detector are mapped to radiance through a calibration coefficient for each wavelength. Thus, the spectral power flux on the projected area can be plotted as a function of wavelength, creating a radiance fingerprint. When exposed to a constant source, the radiance from plant canopies can be converted to hyperspectral reflectance to facilitate the identification and comparison between different scans. The radiance-to-reflectance conversion is done through the use of dark and white references [19,55]. Prior to each scan, the camera when capped captures an internal dark current as the dark reference. The white reference is acquired by a standard 25.4 cm square Labsphere Spectralon® (Labsphere, Inc., North Sutton, NH, USA) made of barium sulfate, which can reflect 99.7% light.

All pixels on the image are subjected to radiometric calibration by subtracting the dark reference and normalized by the white reference before any feature extraction from hyperspectral reflectance curves. All the conversion and correction are done in SpectraView software (Version: 1.1.38, Headwall Photonics, Inc., Bolton, MA, USA). The raw reflectance is normalized as shown in Equation (1) to range from 0 to 1. This transformation makes the spectral signatures between scans comparable [53,56].

$$I_r = \frac{R - D}{W - D} \tag{1}$$

where $R$ and $I_r$ are the raw and normalized reflectance intensities from a target pixel; $D$ and $W$ are its corresponding dark current and white reference, respectively.

### 2.4. Optimization of Raw Reflectance

The spectral curves retrieved from plants are typically not smooth, especially from a single pixel [57]. The Savitzky–Golay smoothing technique provides a moving average of $n$ adjacent bands and fits the averaged points with an mth-order polynomial function. Another smoothing strategy to increase the signal-to-noise ratio (SNR) is to put $l$ neighbor bands into one bin [53,58–60]. Furthermore, a spectral curve is extracted from binning pixels within a spot (on leaf surface) instead of picking an independent pixel. Figure 2 compares the two ways of data extraction: single pixel and spatial binning. It can be seen from Figure 2 that the noise is suppressed along the spectrum, and especially in the VNIR and the SNR, it is almost doubled by the spatial binning. The above raw hyperspectral reflectance curve extraction is achieved in the classification module of SpectraView software.

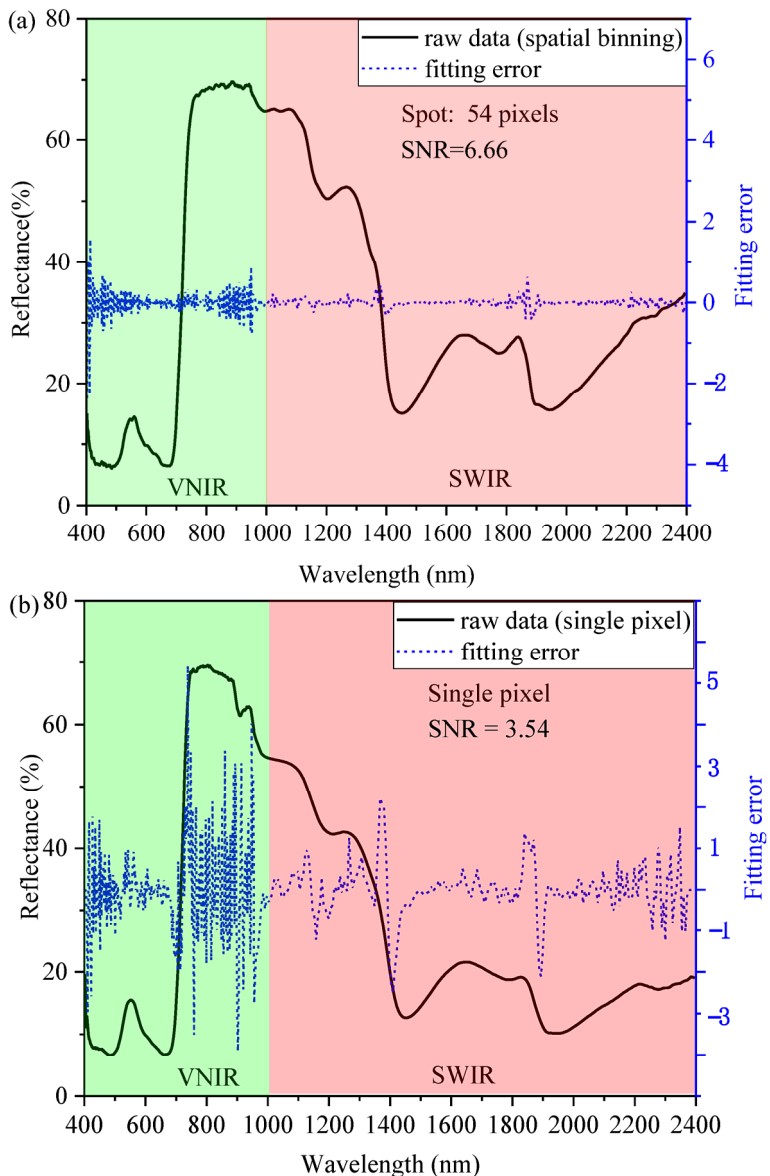

**Figure 2.** SNR of hyperspectral reflectance extracted from two strategies: (**a**) spatial binning of multiple pixels, and (**b**) spectral averaging at a single pixel.

In addition, the canopy leaf inclination and thus distance to the imager detector affect the intensity of reflectance [15,19]. Such variations can lead to locally higher reflection in some pixels and thus neutralize an artifact unintentionally [54]. To compensate for the above multiplicative factors, the standard normal variate (SNV) is introduced to normalize the spectra by subtracting their mean from the raw data and dividing the difference by their standard deviation as indicated in Equation (2) [51].

$$C_{SNV} = \frac{R - mean(R)}{STDEV(R)} \tag{2}$$

where *R* and *C* denote the spectral curve before and after the SNV processing, respectively; *Mean* and *STDEV* are the mean value and standard deviation of a spectral curve, respectively.

Mathematically, SNV can be viewed to rescale the variables such as leaf inclination and light scattering into a standard form. This transformation retains every minor feature of the original spectral signature, thus making the reflectance curves from pixels in the

ROI comparable. After the SNV, the raw hyperspectral reflectance was differentiated concerning wavelength to further reduce the effect of multiplicative variations [58]. More importantly, derivative analysis augments absorption features that are masked by the noise [18,54,61,62]. In this study, both the first-order derivative (FOD) and the second-order derivative (SOD) of each hyperspectral reflectance spectrum are calculated by Savitzky–Golay filtering with a window of 9 bands and a polynomial order of 2. The Savitzky–Golay filtering can simultaneously smooth and differentiate the spectra following a least square optimization [60,61]. All data transformations are done by using the Unscrambler X software (Version: 10.1).

*2.5. Multivariate Analysis*

Each hyperspectral reflectance spectrum contains 489 sampling wavelengths between 400 and 2400 nm. The features manifested within a close range of wavelengths are sometimes interrelated since the chemical stretches interact with photons that have nearly tantamount energy. The reflectance curve also exhibits variations in the VNIR and SWIR range in terms of the importance of various stressors, which camouflages the distinction of each treatment and complicates the stress discrimination. To account for the above factors, multivariate analysis has proven effective in successfully distinguishing stress treatments [39,42,58].

2.5.1. Principal Component Analysis (PCA)

A single spectral curve can be expressed into a function of wavelength as displayed in Equation (3),

$$H(w_1,\, w_2,\, \cdots w_p) \quad = f_{w1}(w_1,\, w_2,\, \cdots w_p) \\ + f_{w2}(w_1,\, w_2,\, \cdots w_p) + \cdots \\ + f_{wp}(w_1,\, w_2,\, \cdots w_p) \tag{3}$$

where $w_i$ represents the wavelength $w$ at the $i$th band out of a total number of samplings $p$ in the spectral range of interest (e.g., 489 samplings of the full spectra); $H$ denotes the spectral curve function; $f_{wi}$ is the integration of contributions from $p$ number of wavelengths centered at the $i$th wavelength (i = 1, 2, ..., p). Multicollinearity at adjacent wavelengths often influences the interpretation of spectral curve features due to high dimensionality and entangled correlation. For the construction of a discrimination model, all features of the hyperspectral reflectance curve from each stress treatment can be expressed into $X_s = [H_0,\, H_1, \cdots, H_{s-1}]$, where $s$ is the number of observations. $H_0$ is a collection of the p terms in Equation (3) and is written in vector form. To reduce the dimensionality of hyperspectral data, PCA is used to analyze the original and transformed spectral curves. PCA projects the p-dimensional data points to different orthogonal axes by maximizing the variance in each direction. The PCA process makes the information along principal component (PC) axes independent. The projection space is defined by the eigenvectors that are derived from matrix $X_s$. The eigenvalues of the directions associated with the eigenvectors can be computed from Equation (4) [63]. The top eigenvalues are the indicators of the explanatory variance of the original data in the corresponding PC directions.

$$E^{-1}CE = \lambda \tag{4}$$

where $C$ denotes the covariance matrix of the $X$; $E$ is a matrix that represents a collection of the computed eigenvectors; and $\lambda$ is a matrix including the eigenvalues in diagonal direction. The eigenvalues are displayed in descending order and a higher eigenvalue means more contribution of that PC to representing the original data. The number of PCs, $n$, to use for further model construction is determined by a 95% accumulative explained variance of the raw reflectance data by the first $n$ PCs. While increasing $n$ unproportionally can cover more radiance, an excessive number of PCs is likely to cause dimensionality havoc or Hughes phenomenon [64]. After the PCA, the features of the $X_s$ can be remodeled as $F$ with the $n$ eigenvectors.

$$F = X_s E_n \tag{5}$$

### 2.5.2. Linear/Quadratic Discriminant Analysis (LDA/QDA)

The optimized and transformed raw reflectance data with reduced dimensions will be used to identify or distinguish various treatments among three species of plants. LDA/QDA groups different stressors by modeling the difference among samples through the feature vectors *F* from the PCA analysis. QDA is performed by projecting features to hyperplanes that maximize the distances between categories and minimize the variation within each category. Mathematically, QDA classifies samples by maximizing the ratio between $C_w$ (within-group covariance) and $C_b$ (between-group covariance) [63]. Once the maximum ratio is located, the optimal differentiation space $S_{QDA}$ is expressed into

$$S_{QDA} = \arg\, max_{QDA} \frac{\left|S^T C_b S\right|}{\left|S^T C_w S\right|} \tag{6}$$

where $S_{QDA}$ represents the direction in which the groups are sectioned optimally in feature data space *F*. LDA is a simplified form (special case) of QDA in terms of discrimination strategy. LDA groups two stresses with a linear boundary while QDA can section more stressors with multiple quadratic boundaries. A lower-class discrimination with QDA risks overfitting. A higher-class LDA is likely to fail in classification due to the limited linear separability [38,42].

Five treatments (gas leakage, SI, HMC, DE, and a control group) were discriminated against with the QDA. The differentiation of stressors was carried out on any individual group of plant species. To do so, five leaves were randomly selected from the canopy of each plant for spectra collection. For an individual leaf, the representative spectrum is obtained by averaging spectra from three different spots within the domain of the leaf. Hence, a total of 15 spectra are accumulated from each scanning. Overall, the stress treatment lasted 60 days in this study. The discriminant analysis used the hyperspectral data from all the collection dates and the spectra were divided into 70% and 30% for model training and testing. The classification is accomplished with the Unscrambler X software (Version: 10.1). The flow chart of the hyperspectral reflectance collection, processing, and classification is presented in Figure 3. The entire process is divided into three steps: data acquisition, data transformation, and model construction.

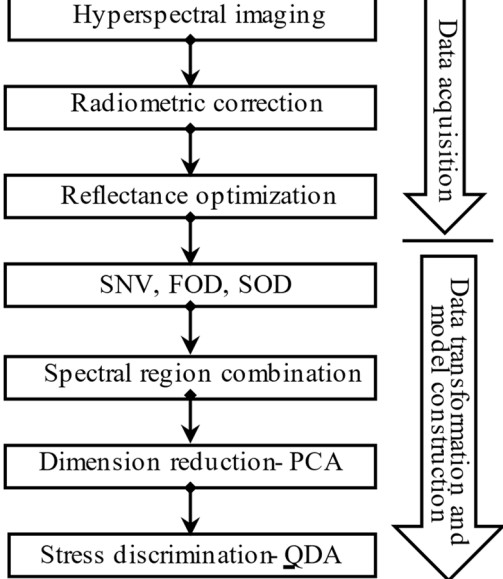

**Figure 3.** Flow chart of hyperspectral reflectance collection, processing, and stress discrimination.

## 3. Results

### 3.1. Raw Hyperspectral Reflectance Extraction and Stress Symptoms

The hyperspectral responses under non-optimal ambient stress conditions were considered different from those under optimal/normal conditions due to the foliage composition and morphology changes of plants [44]. The plant stressors were reported to be sensitive to the red edge position (REP), which is highly correlated with the chlorophyll content in leaves. The corresponding red edge position indexing (REPI) was mapped over the plant canopy in the ENVI software (Version: 5.5.2.), as shown in Figure 4a,b, and the development of each stressor was plotted in Figure 4c. The REPI values for Grass, Gem, and South are approximately 0.713, 0.725, and 0.709, respectively. Gem yields the highest REPI because the thick shrub leaves contain more chlorophyll. After 25 days of treatment, it is observed that the plant stressors cause slight changes in REPI when compared with the plant in the control group. The changes spread more widely when the plants are treated for 60 days. These changes are indicative of accumulative stress effects on the plants. However, the stress did not necessarily yield visible symptoms on the treated plants during the 60-day test. Among all three species, shrub South was the most sensitive to gas exposures and displayed chlorosis on all the stress treatment scenarios due to the reduced photosynthesis, as shown in Figure 5. The earliest chlorosis appeared on the plants with HMC and DE treatments. For DE, the loss of sheen on the plant leaf was another notable feature because of the dehydration effect. At the last stage of the stress development, some of the South leaves demonstrated a dark/brown rim, which was reported as a sign of necrosis. Grass also showed a sign of chlorosis under the stress conditions and started from the tip of the long leaves. Compared to the Grass, the Shrub South showed less visible symptoms and the Shrub Gem did not show identifiable physical changes on leaves regardless of the stress treatments though optical differences can be observed in Figure 4(c-2). The resistance of the stress effect likely accounts for the difference among the plant species. As such, separate discriminant analyses are performed on the three plant species.

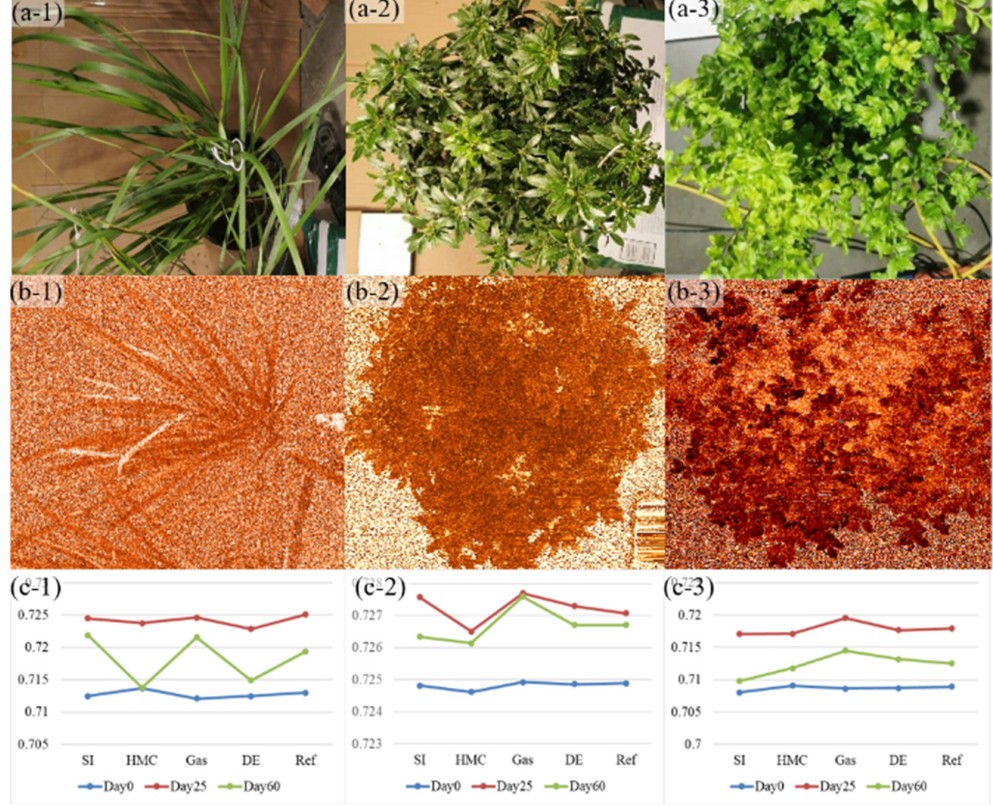

**Figure 4.** Demonstration of (**a-1**–**a-3**) plants, (**b-1**–**b-3**) mapping of REPI, and (**c-1**–**c-3**) change of REPI among stress treatments.

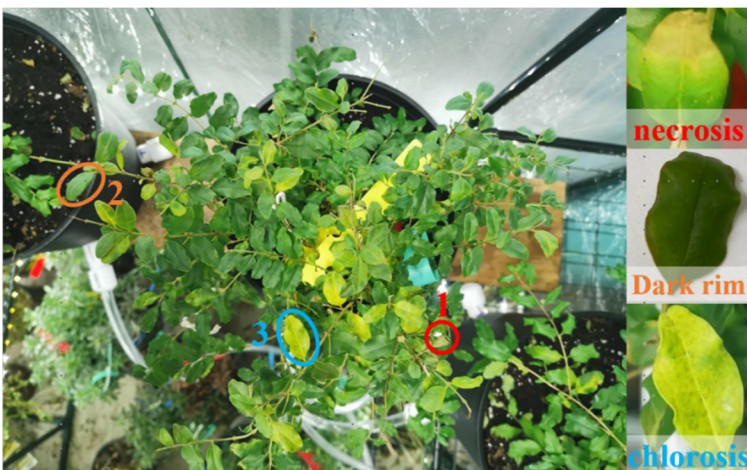

**Figure 5.** Visible stress symptoms in the Shrub South.

*3.2. Spectral Correlation and Dimensionality Reduction with PCA*

Hyperspectral redundancy, which is indicative of the similarity between bands, was quantified by the spectral correlation. Figure 6 shows the coefficient of correlation between any two bands in VNIR and SWIR ranges, respectively for the test plant. In the VNIR range, the most informative wavelengths are located around 540 nm in green and 680 nm near red. The 540 nm band represents a featured absorption of chlorophyll, and 680 nm is also dominated by the presence of chlorophyll [65]. The NIR spectra are regulated by the scattering effect of the cellular components and thus are less correlated with the pigment absorptions in VIS. In the SWIR range, the spectral information between 1000 nm and 2000 nm generally represents water features because the water or water-related content yields spectral features at 1110 nm, 1380 nm, 1430 nm, 1780 nm, and 1920 nm [66]. Therefore, the spectral bands are correlated as indicated by the high Pearson coefficient values. The reflectance at bands between 1980 nm and 2400 nm is related to the plant biomass such as lignin and cellulose and thus not correlated with information at the lower wavelengths [15]. Those informative wavelengths were also demonstrated by a band redundancy study [39].

Based on the understanding of electromagnetic radiation and the interrelation of information at various wavelengths, the original hyperspectral data were compressed for efficient and effective stress detection. Figure 7 displays the score plots of all PCs for the classification model of Ref-Gas-DE using different spectral transformations and Figure 8 demonstrates their corresponding spectral loadings. The original 249 bands in the VNIR range can be reduced to three in all transformation scenarios. As indicated by Figure 7, the first two PCs explain more than 90% of the original spectra information and the first three PCs represent over 95% of the variance of spectra in the original space. By examining the spectral loadings of the corresponding PCs, the most informative bands are located in VIS (400–680 nm) and 'red edge' (680–740 nm) in both raw and SNV transformed spectra. The secondary wavelengths that contain significant variances are in the NIR (700–1000 nm) range. It agrees with previous findings in the research on plant stress identification and detection [33,38]. As for the effect of derivative analysis on the PCA model, the 'red edge' ranks the most significant region and the information in VIS contributes comparatively less, as indicated by the spectral loading of PC1 in Figure 8c. Similarly, based on the PC2 and PC3 contributions, only some pigments' featured absorption bands are highlighted, such as 560 nm (chlorophyll a), and 610 nm, and 670 nm (chlorophyll concentration). The informativeness of 'REP' was demonstrated by the zero point in PC3 in correspondence to the 'red edge' in PC1. The effectiveness of the REP was reported to shift to lower wavelengths in the presence of stress in many studies [33,35]. In comparison with raw spectra, the SOD is less effective than the FOD in the PCA analysis. Overall, four PCs are needed to explain the 95% variance, and only the 'red edge' is identified, which has a trivial effect on the dimensionality reduction of the raw spectra according to the loadings. Instead,

the wavelengths at the beginning of the VIS (400–450 nm) include most contributions from the perspective of variance. The reason is that the SOD amplifies the noise in the region, therefore inducing more variance.

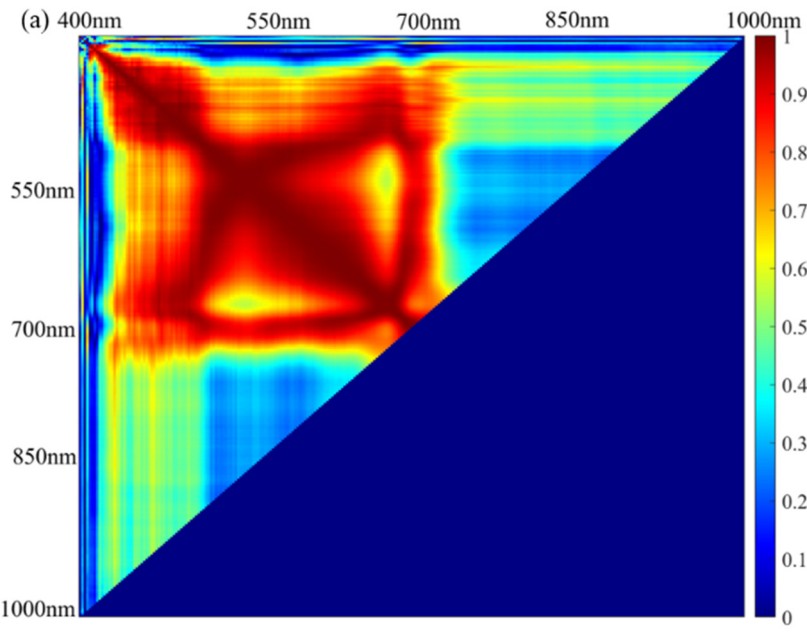

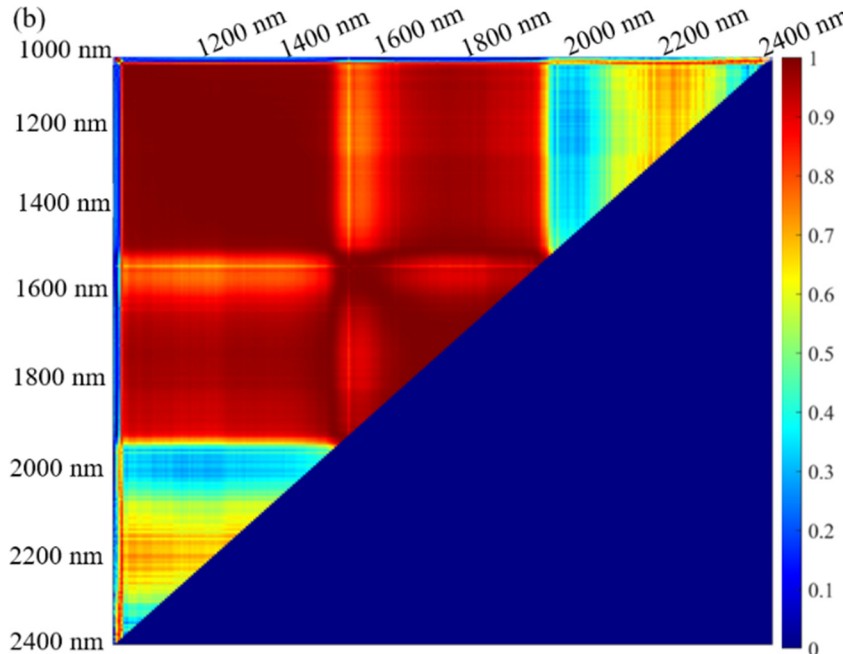

**Figure 6.** Pearson correlation coefficient of wavelengths obtained from the hyperspectral imaging on vegetation leaf in the range of: (**a**) VNIR and (**b**) SWIR.

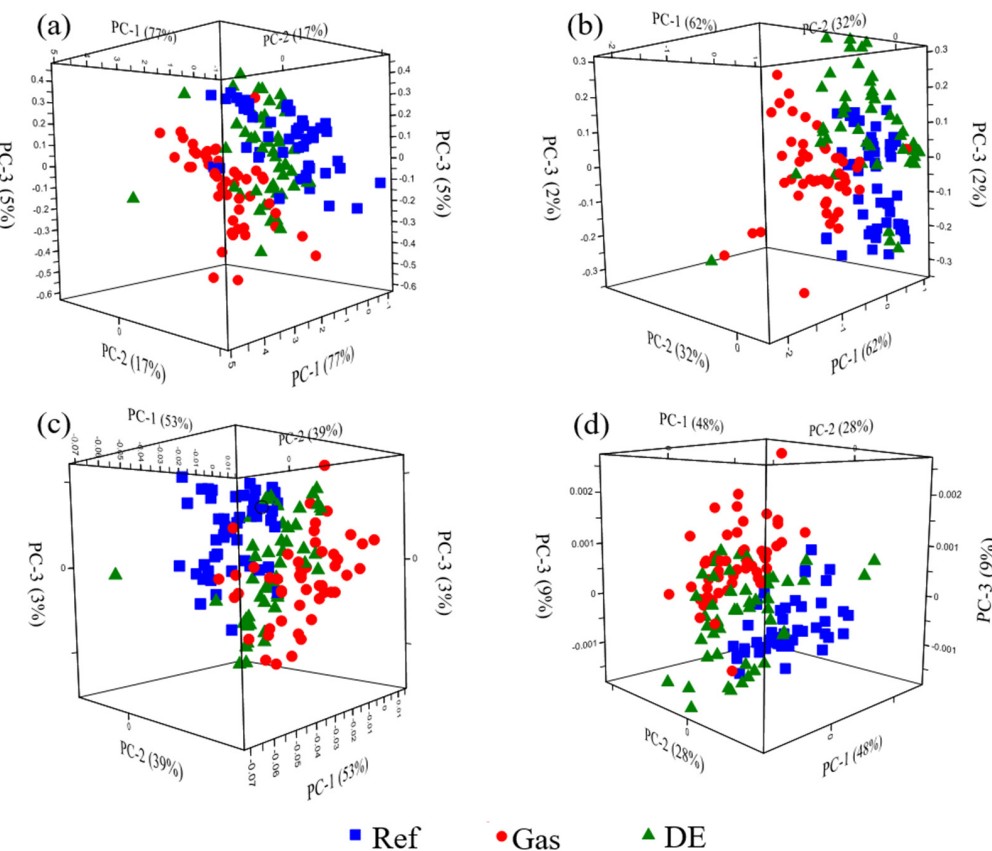

**Figure 7.** PCA score plots in Ref + Gas + DE derived from data based on (**a**) raw spectra, (**b**) SNV spectra, (**c**) FOD spectra, and (**d**) SOD spectra.

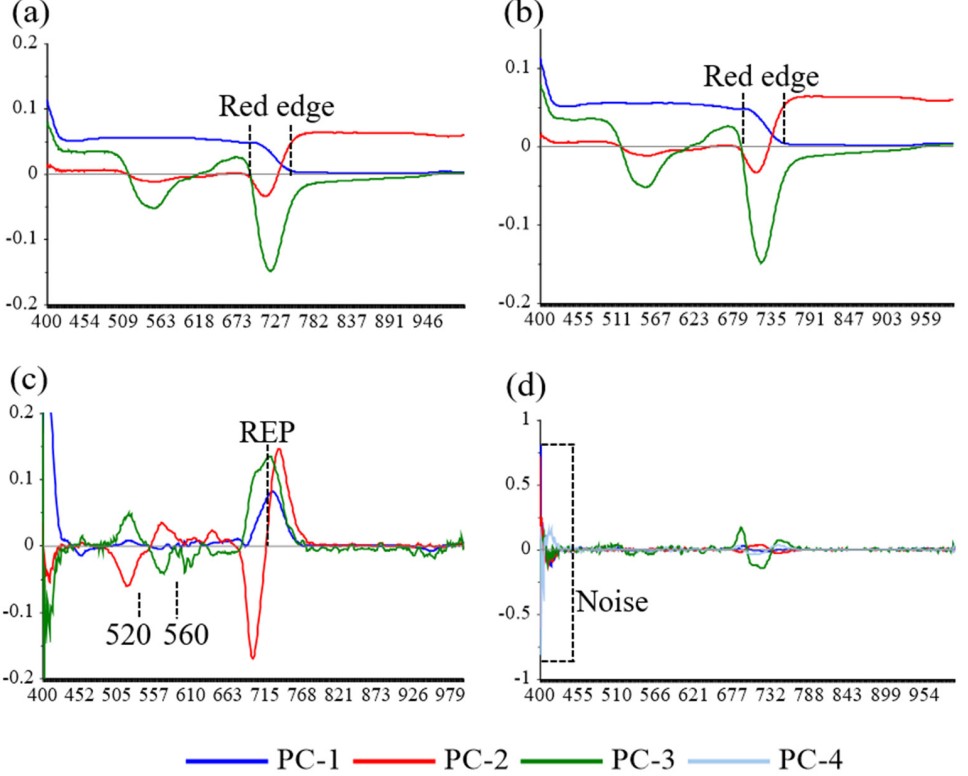

**Figure 8.** Spectral loadings of PCs in Ref + Gas + DE derived from data based on (**a**) raw spectra, (**b**) SNV spectra, (**c**) FOD spectra, and (**d**) SOD spectra.

### 3.3. Gas Treatment Identification with LDA

The VNIR, SWIR, and full spectrum were considered respectively in the LDA model for gas stress identification. Only the hyperspectral reflectance curves acquired from the gas-stressed and untreated control sets of plants were pooled in this case. Figure 9 presents the gas stress identification results with LDA for different plant species and the classification accuracies of Ref and Gas are indicated by blue and orange color bars, respectively. The identification accuracy shows a clear difference between plant species. The Shrub South yielded the most accurate result while the Shrub Gem was the least. The overall average identification accuracy is 79.3% for the South, which is approximately 10% higher than that of the Gem. It is probably because the stress in the Gem was developed much more slowly than that in the other two species. Therefore, many unstressed samples in the early stage are eventually labeled as gas-stressed during the entire acquisition period, which will provide erroneous data in the development of the LDA model. The small changes in REPI due to stress treatment, as shown in Figure 4(c-2), may also explain the delayed stress development in the Gem.

The spectral range selection also affects the identification of the gas stress on plants by LDAs. The most informative gas stress identification comes from spectra in the VNIR range for all three plants. The SWIR range provides the least information since it is less responsive to the methane gas stress, as reported in [67]. The full spectra from 400 nm to 2500 nm yield intermediate accuracy because the SWIR range can influence the LDA model in identifying the true gas stress. In addition, the gas stress identification accuracy significantly differs between the spectral transformations. LDA models based on the spectra in the VNIR range after the FOD transformation generally yield the most desirable gas stress and non-stress classifications. The highest identification accuracy is 80.2%, 77.6%, and 86.8% for the Grass, the Gem, and the South, respectively. The effectiveness of the FOD of original spectra for stress identification was also documented in previous publications [19,42]. Compared to the SNV, the FOD helps reduce the scattering effects from the environment. The FOD analysis exposes some important spectral features such as the red edge position (REP). That is why the identification accuracy after the SNV processing is not as good as the FOD even after the ambience effects have been removed. The SOD also reduces the multiplicative effect in the process of differentiation. However, the higher-order derivative analysis would augment the noise in the original spectra, especially at the beginning and ending region of the spectra as shown in Figure 4(c-3). The noisy part in the SOD space is predominant compared to other spectral features. When taking the varying sources into account in control and gas plant classification, the LDA model can mispresent the actual scenario.

### 3.4. Gas Stress Discrimination from Other Treatments with QDA in Multi-Class Classification

SI, HMC, and DE were considered as distraction parameters for the detection of gas leakage in multi-class discrimination. Considering different covariances between classes, QDA was used to find optimal boundaries of the classes with a nonlinear function [37,56,58]. The hyperspectral curves collected from three species of plant under five treatment conditions were pooled for QDA classification.

Figure 10 presents the gas stress classification accuracy with the other plant stressors considered in this study. The classification accuracy of each stress effect was indicated. The highest gas classification accuracy is 64.2%, 61.3%, and 68.7% for Grass, Gem and South, respectively. Among all the five classes, the shrub Gem still yielded the lowest accuracy. Plant species make a huge difference in both the gas stress identification as discussed in Section 3.3 and classification because the plants vary significantly in stress resistance, pigment concentration, and root structure [38]. Therefore, plants do not show synchronized responses even when exposed to similar stress stimuli. In addition, the classification accuracy of each stressor varies significantly between the spectral bands used. SWIR still produced the worst results and VNIR and full spectra both yielded higher accuracy in all five stressor classifications. VNIR spectra performed slightly better than the full spectra. The weaker SWIR was likely related to the slow psychological response of the

plant because the informative bands in SWIR were regulated by the biomass components of the plant, which are reluctant to alter compared to the plant-pigmentation-dominated bands in the visible range [68].

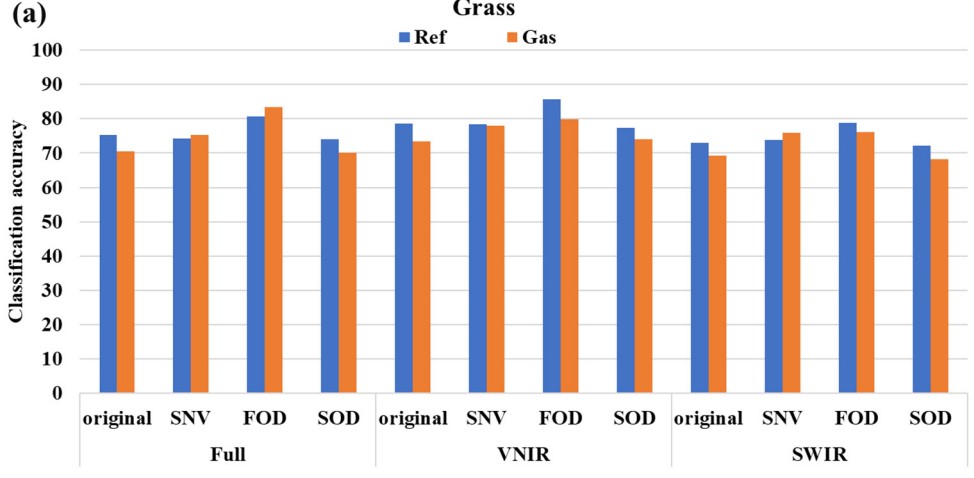

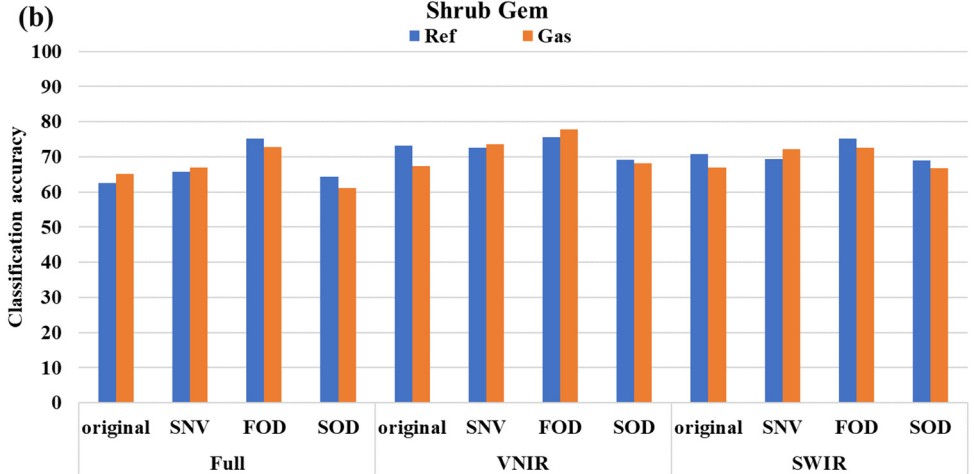

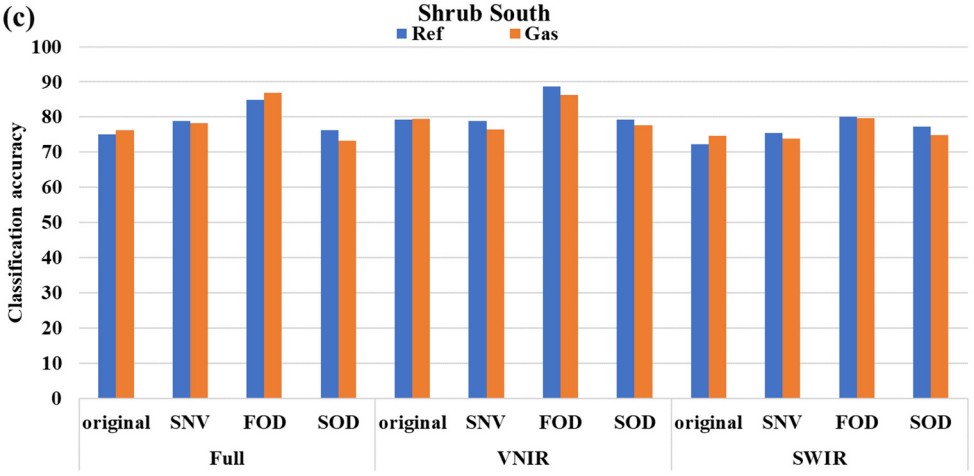

**Figure 9.** LDA-based gas stress identification from different plants: (**a**) Grass, (**b**) Shrub Gem, and (**c**) Shrub South.

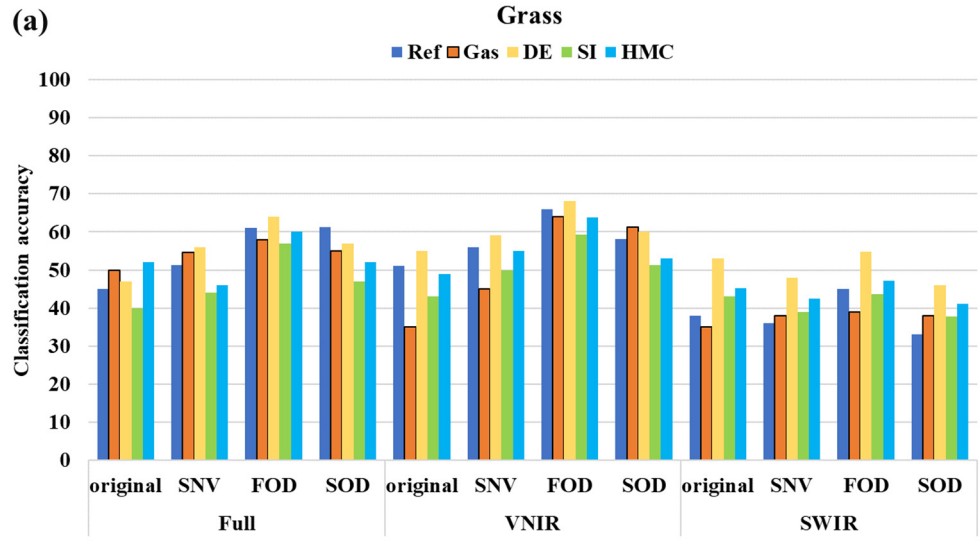

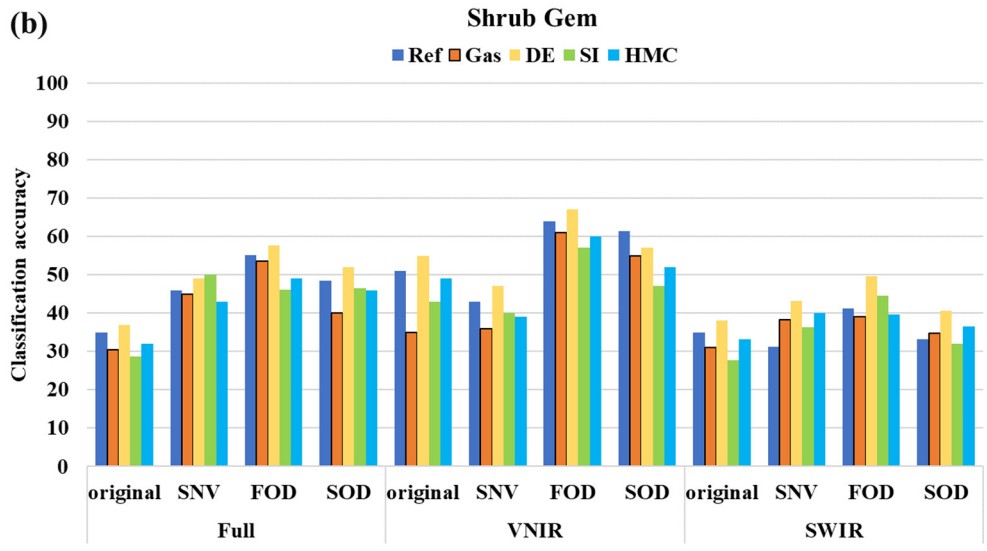

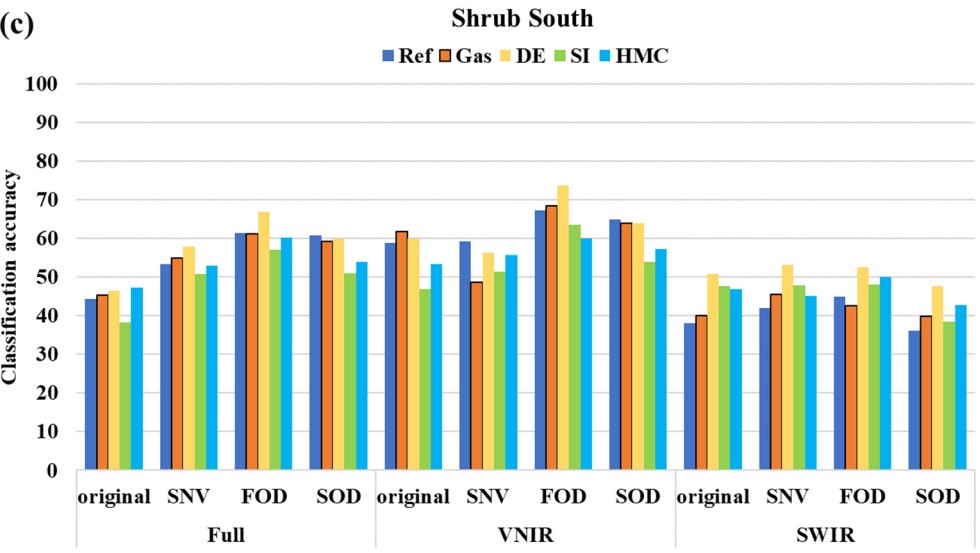

**Figure 10.** QDA-based gas stress discrimination from three other stressors on different plants: (**a**) Grass, (**b**) Shrub Gem, and (**c**) Shrub South.

Looking into the accuracy difference between the plant stressors, it is noted that DE can be more accurately classified by the QDA models from all three plant species. The best DE classification rate is 74.2%, which was derived from the South with VNIR data after the FOD analysis. In addition, the DE is also highly detectable compared to the rest of the stressors with the spectral information in the SWIR range. This is because the DE treatment significantly reduced the water content in the leaf tissues. Loss of hydration vastly increased the reflectance in the SWIR range [19]. The reduction of the leaf water also increased the reflectance in the visible range due to the decline of the chlorophyll content as observed from the Z. Mays barley [69,70]. Moreover, plant photosynthesis activities were prohibited to some extent due to the lack of water supply. Hence, the optical properties in the VNIR range change correspondingly. In contrast, SI achieved the lowest accuracy in most classification scenarios. It is probably attributable to the minor change in reflectance due to the saline effect. The SI was reported to increase the NIR reflectance on the plant leave scale [38]. The VIS range may not change because the SI does not directly reduce the concentration of the pigment and thus yields minor differences in the visible range (400–600 nm). However, HMC exerts the stress influence directly in the biochemical way because the HMC reduces the chlorophyll in plants. The center Magnesium can be replaced by the other heavy metal cations; thus, the photosynthesis can be interrupted because the photons cannot be effectively captured. The reactive species produced during photosynthesis cannot be compromised to generate necrosis on the leaf surface. As a result, the VNIR spectra adapts to the biochemical transformation that differs from the effects of the remaining stressors. That explains why the HMC can also be discriminated from gas with acceptable accuracy.

### 3.5. Gas Stress Discrimination from Another Treatment with QDA in Three-Class Classification

Section 3.3 and Section 3.4 discussed two extreme scenarios for gas stress detection with no and three disturbances, respectively. In applications, gas leakage in an environment with one disturbance stressor is more practical since multiple natural stressors (SI, HMC, and DE) are seldom present at one location. Sections 3.3 and 3.4 also indicated that the hyperspectral reflectance in the VNIR range can yield the most accurate detection results. Thus, only spectra in the VNIR range are included in the following analysis. Figure 11 shows the gas stress detection results with QDA when the gas stressor is pooled with Ref (stress-free) and one other predefined stressor on three species of plants. In Figure 11, 'Ref-Gas-DE' denotes a hyperspectral reflectance pool of three treatment conditions: no stress reference, gas treatment, and drought exposure.

Among the three natural plant stressors, DE distinguishes itself from the others in spectral reflectance and thus yields the highest classification accuracy in the five-class classification in Section 3.4. In the current three-class classification, the DE also achieved the highest accuracy. The shrub South reveals the most accurate stress classification, which reached 88.2% by using the FOD transformed spectra. One of the reasons for the easy discrimination of DE is that the DE yields very distinct spectral profiles against the Ref and Gas, as discussed before. Figure 11 shows the representative spectra retrieved from the plants after 60 days of stress treatment. REP was shifted to the lower wavelengths for approximately 9 nm as opposed to Ref. The blue shift was also noted in the DE research and other plant stress detection studies [21,38]. For the methane gas stress, Nooman and Smith (2008) also detected the REP blue shift, though only 1–2 nm [71]. In comparison, the gas-induced REP shift in this study is approximately 3 nm, as indicated by the FOD spectra in Figure 12. The SI remains the most difficult stressor to distinguish from the gas from the perspectives of the QDA model. With the presence of the SI, the highest gas identification accuracy is 81.3%, which is approximately 3% lower than the classification combination of Ref-Gas-HMC. The confusion in classification may result from the close hyperspectral reflectance, as both stressors were reported to not alter the optical properties in the VIS and only slightly lower the reflectance in the NIR [72,73].

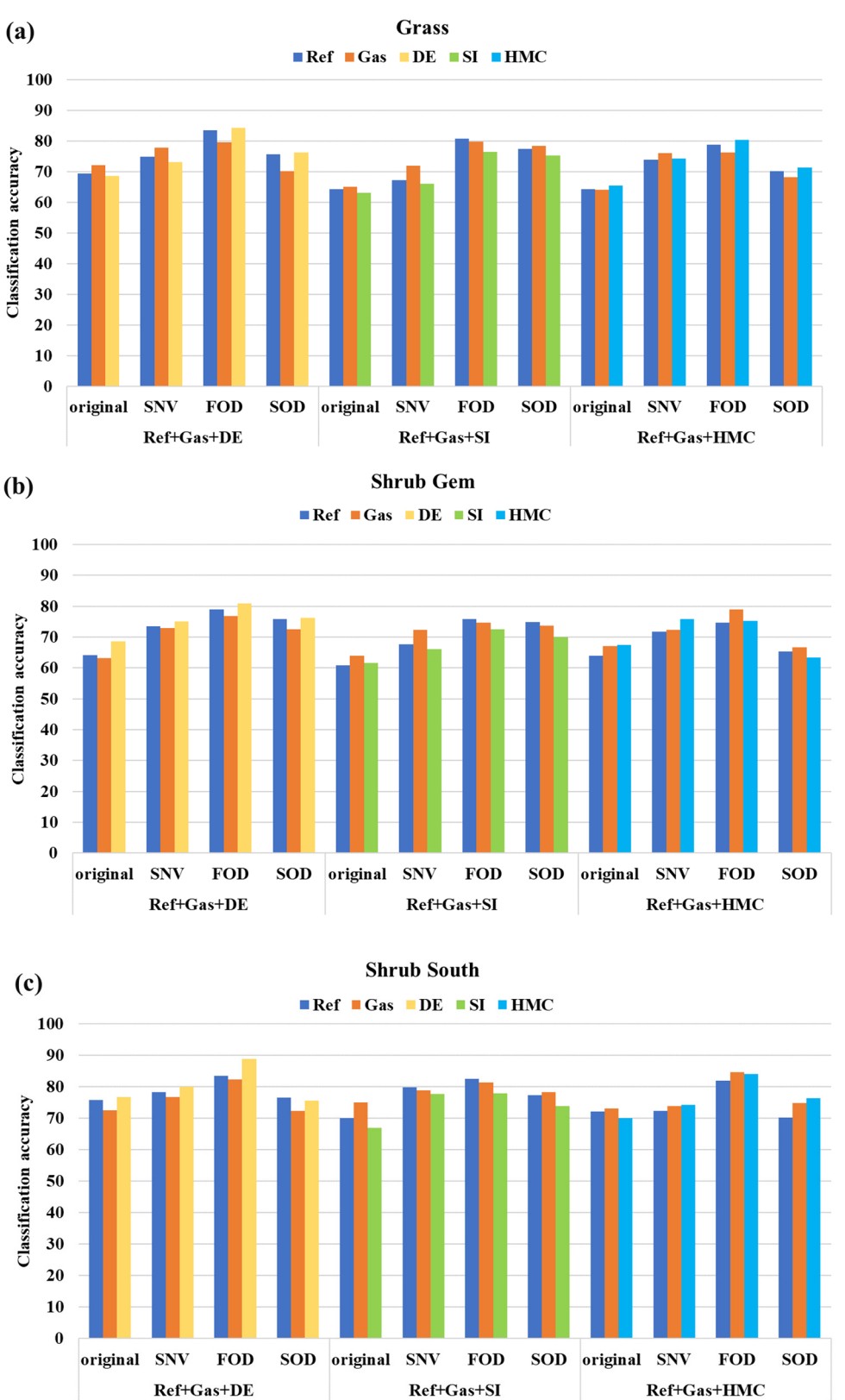

**Figure 11.** QDA of VNIR-ranged spectra for gas stress discrimination from one distraction stressor on different plants: (**a**) Grass, (**b**) Shrub Gem, and (**c**) Shrub South.

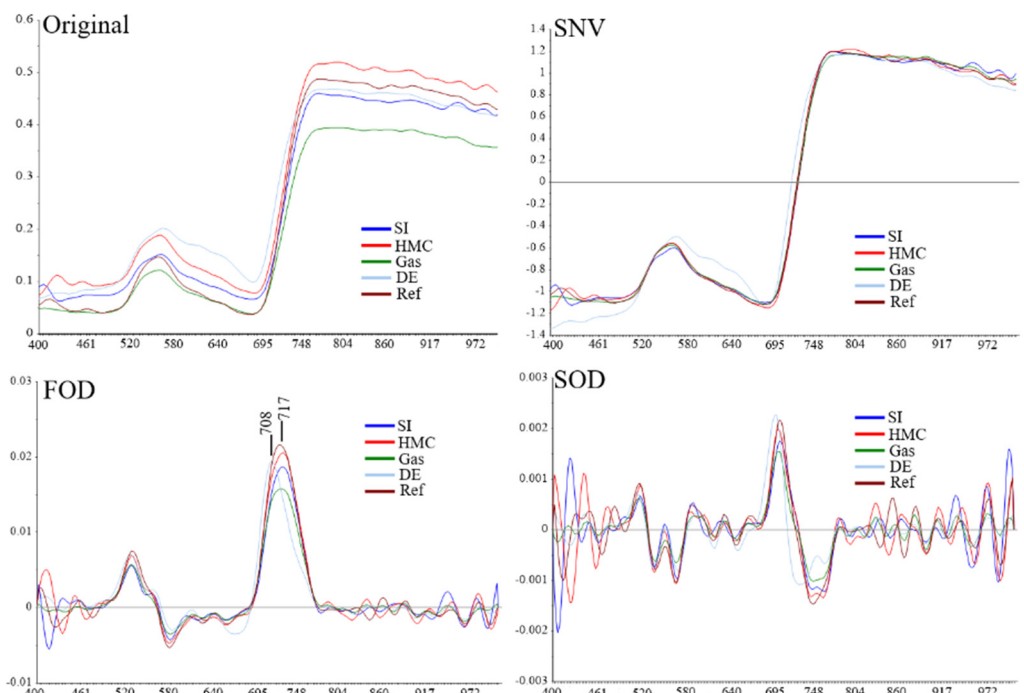

**Figure 12.** Representative spectra of the various transformations under five stress treatments on the shrub South at Day 60.

Among various transformations of the original spectra, SNV sometimes does not influence the classification results, though it can remove the environmental disturbance, which was also reported in [19]. The FOD transformation improves the contrast of spectra associated with different treatments by reducing the multiplicate effect. In addition, the normalization process suppresses many non-informative bands while maintaining significant absorption features in the original space. In contrast, SOD yields the lowest classification results in most scenarios because the SOD makes the noise even more prominent than the original signal to retard the classification. Moreover, the higher order derivative analysis disperses the noise that will contaminate more bands in the VNIR range and thus make it more difficult in classification.

## 4. Discussion

Natural gas pipelines account for 80–90% of the transmission pipeline network, spanning approximately 3 million miles across the US [2]. Annually, these pipelines emit around 200 million cubic feet of natural gas, leading to significant environmental impact and substantial financial losses [66]. Studies by Naga Venkata et al. [74] and Lu et al. [3] have delved into the progress, challenges, and opportunities in detecting pipeline leaks using various techniques. However, achieving affordable and effective early detection of underground natural gas leaks remains a formidable challenge. This article proposes leveraging ground vegetation to detect underground natural gas leaks, employing hyperspectral imaging to monitor biochemical changes in spectral profiles that signify the presence of a leak. Specifically, this study tested the hypothesis that vegetation can be responsive to the natural gas impact. Moreover, this study investigated the discrimination of other vegetation stressors in the presence of natural gas by considering a more complicated real-world natural gas leak detection scenario and enhancing the robustness of the proposed method.

Hyperspectral imaging of the vegetation was utilized to perform the natural gas leak detection and discrimination. In the context of early detection of natural gas leak, it is postulated that stimuli in this study were not necessary to alter the spectral profile in the full range. Therefore, three spectral zones were considered in this study: visible near infrared (VNIR), short-wave infrared (SWIR), and the full spectra. The results indicated

that VNIR spectroscopy consistently yields higher accuracy in natural gas identification and discrimination. This is because plants tend to call for various pigments to counteract the immediate adverse effects [75,76]. Principal Component Analysis (PCA) indicates that wavelengths associated with the red edge and chlorophyll contribute significantly according to the larger spectral loadings across all PCs in the natural gas discrimination study, highlighting plant pigment vulnerability to vegetation stressors in the presence of vegetation stressors [37,53,77]. This study emphasizes the sensitivity of different spectral zones to use hyperspectral imaging to reflect vegetation stress in the presence of other stressors. The inclusion of nonresponsive or less active wavelengths (e.g., the full spectra) in the model complicates the natural gas detection, thus potentially impairing the detection accuracy. In addition, it is worth noting that spectral transformation also plays a part in the detection accuracy. First-order derivative (FOD) consistently facilitates natural gas detection and discrimination. Even in the VNIR range, there remain some nonactive bands, and FOD can basically eliminate their disturbance on the natural gas leak detection. However, the second-order derivative (SOD) presents the least detection accuracy because SOD amplifies the noises in the original spectral and even obscures the useful information as indicated in Figure 12.

In the natural gas leak identification and discrimination, the results indicated that vegetation can be used as a medium to receive the natural gas impact and the alterations on vegetation can be identifiable in spectra, as demonstrated by the high detection accuracy. As illustrated in the previous studies, many vegetation stressors, both biotic and abiotic, such as saline soil, water deficit, and heavy metal contamination also induce spectral changes and can be identifiable in hyperspectral imaging, which might increase the false alarms in the natural leak detection via vegetations [5,21,40,53,72,73]. In the presence of distraction vegetation stressors, natural gas stress remains discernible though with a lower accuracy in this study. It is noticed that the drought differs from natural gas the most on vegetation spectra as illustrated by the highest discrimination accuracy across plant species. It might result from the distinct spectral responses under natural gas and drought effects. It is reported that the drought exposure increases the hyperspectral reflectance in the range of NIR while it decreases in 450–530 nm; the trend is opposite in the presence of drought [31,32,75,76]. The difference in the spectral response comes from the generic activities under two stressors [33,75,78,79]. In contrast, saline soil might confound the natural gas leak detection, as indicated by the lowest discrimination accuracy among the three distraction stressors [53,72]. Such confusion in natural gas detection comes from the similar spectral response because salinity also decreases the reflectance in the NIR as of natural gas, which is consistent with the results from the authors' field natural gas leak detection study.

Though the natural gas leak can be identified and differentiated via vegetation, the generic reasons and the consequential biochemical products on vegetation behind the natural gas stress remain insufficiently investigated. Future endeavors can be allocated to biochemical studies to reveal the stress mechanisms to support the interpretation of the hyperspectral data. In addition, with the proven capability of natural gas detection through vegetation in a lab environment, it is necessary to test the applicability of deploying hyperspectral imaging in the field of natural gas leak detection.

## 5. Conclusions

This paper summarized the feasibility study on detecting methane gas stress on vegetation from hyperspectral reflectance as it contains variance of the vegetation derived from exposure to the stress. Due to plant generic responses to electromagnetic radiation, the transformed hyperspectral data in different spectral ranges (VNIR, SWIR, and full spectra) were compared for the first time. The multivariate analysis technique (LDA or QDA) was used to statistically differentiate the gas stress from both the unstressed reference and three natural stressors (DE, HMC, and SI). Based on the extensive tests, data normalization analyses, and noise cleansing, the following conclusions can be drawn:

1.  The LDA can be applied to effectively identify the gas stress on vegetation from unstressed vegetation with an accuracy of 77.6–84.8% in the two-class detection process.
2.  With the distraction of three natural stressors, the QDA can be applied to discriminate the gas treatment from natural stressors with an accuracy of 61.2–68.4% in the five-class detection process. DE is the most distinguishable stressor, while SI is the least in terms of classification accuracy.
3.  When distracted by one natural stressor (DE, HMC, or SI), the QDA can differentiate the gas stress from the distracted natural stressor and the unstressed reference with an accuracy of 76.4–84.6% in the three-class detection process. This level of accuracy is comparable to that of gas stress identification from unstressed vegetation. These results have practical implications for the natural gas and oil pipeline industries.
4.  The FOD of the VNIR-ranged spectra (400–1000 nm) can always lead to the highest accuracy in almost all detection cases. The FOD can effectively simplify the feature space of raw data by reducing the number of PCs required for more accurate classification.

Overall, the proposed LDA and QDA can be respectively applied to the FOD of the VNIR-ranged spectra for the effective identification of gas stress from unstressed vegetations and successful discrimination of gas stress from one other natural stressor. Even so, further research is required to understand the physiological and biochemical alteration of stressed vegetations to develop more explainable distinctions among hyperspectral reflectance curves associated with different stressor effects. Moreover, the applicability of the LDA and QDA in the pipeline industries must be tested over seasons in an experimental field station to quantify the time effect on the accuracy of the algorithms in gas detection.

**Author Contributions:** Conceptualization, G.C. and P.M.; methodology, G.C. and P.M.; software, P.M.; validation, G.C. and P.M.; formal analysis, G.C. and P.M.; investigation, G.C. and P.M.; data curation, Y.Z. and P.M.; writing—original draft preparation, G.C. and P.M.; writing—review and editing, G.C., J.G.B. and P.M.; visualization, P.M.; supervision, G.C.; project administration, G.C. and P.M.; funding acquisition, G.C. All authors have read and agreed to the published version of the manuscript.

**Funding:** Financial support to complete this study was provided by the U.S. Department of Transportation (USDOT) Pipeline and Hazardous Materials Safety Administration (PHMSA) under Grant No. 693JK31950005CAAP at Missouri University of Science and Technology. The views, opinions, findings, and conclusions reflected in this publication are solely those of the authors and do not represent the official policy or position of the USDOT/PHMSA, or any State or other entity.

**Data Availability Statement:** The data in this study is available by contacting the corresponding author.

**Conflicts of Interest:** The authors declare no conflict of interest.

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
