# Peer review of "Natural Gas Induced Vegetation Stress Identification and Discrimination from Hyperspectral Imaging for Pipeline Leakage Detection"

_remotesensing, doi:10.3390/rs16061029_

Round 1

Reviewer 1 Report

Comments and Suggestions for Authors

Author Response

Dear Reviewer,

The reply is contacined in the attachment. 

Best,

Authors

Reviewer 2 Report

Comments and Suggestions for Authors
  1. 1. How do hyperspectral curves respond to spectral features at unique wavelength bands for different stressors?
    2. Do plants display significantly different sensitivities to various stressors in the visible (VIS), near-infrared (NIR), and shortwave infrared (SWIR) ranges during the identification and discrimination of gas-stressed vegetation?
    3. Are plant responses, such as metabolic responses, similar across different species, allowing for the use of specific vegetative indices for multiple sites and vegetations?
    4. Is the selected plant sensitive to natural gas? Additionally, when transplanted into a greenhouse and tested after a period of time, the environment may become more consistent, thereby reducing sensitivity or differences. Have considerations or attempts been made to directly conduct testing without transplantation, and is the effectiveness the same?

  2. 5. While the literature mentions UAVs, it appears that the article primarily focuses on laboratory testing and analysis. If UAVs are employed for relevant testing, additional coordination may be required (recordkeeping, analysis, on-site calibration, flight altitude, resolution, etc.). How should these factors be considered to obtain usable results?

Author Response

Dear Reviewer,

Thanks for your comments. The reply to the comments is contained in the attachment.

Best,

Authors

Reviewer 3 Report

Comments and Suggestions for Authors

The article showcases the use of hyperspectral reflectance to detect stress in vegetation caused by natural gas leaks. This approach is significant as natural gas leaks can have detrimental environmental effects, and early detection is crucial for damage mitigation. The study employs controlled greenhouse experiments using three different plant species exposed to various stressors including natural gas, salinity, heavy metals, and drought to simulate real-world conditions. By analyzing the hyperspectral reflectance data, the research team developed a method to not only detect stress in vegetation but also to distinguish between stress caused by natural gas leaks and other environmental factors. 

Experimental data was tested using both Linear Discriminant Analysis (LDA) and Quadratic Discriminant Analysis (QDA). LDA was used to model the difference between the hyperspectral reflectance curves of gas-stressed and control plants, while QDA was employed to distinguish gas stress from other stressors like salinity, heavy metals, and drought, considering the different covariances between the classes. These tests were applied to the data derived from hyperspectral images of plants subjected to various stress conditions in a controlled environment.

Author Response

Dear Reviewer,

Thanks for your recognition of the author's work. The authors really appreciate for your comments.

Best,

Authors